# Current Immunological and Clinical Perspective on Vaccinations in Multiple Sclerosis Patients: Are They Safe after All?

**DOI:** 10.3390/ijms22083859

**Published:** 2021-04-08

**Authors:** Shani Witman Tsur, Eli Adrian Zaher, Meydan Tsur, Karolina Kania, Alicja Kalinowska-Łyszczarz

**Affiliations:** 1Center for Medical Education in English, Poznan University of Medical Sciences (PUMS), 61-701 Poznań, Poland; shaniwitman@gmail.com (S.W.T.); eliadrianzaher96@gmail.com (E.A.Z.); meydant1@gmail.com (M.T.); 2Department of Neurology, Poznan University of Medical Sciences (PUMS), 60-355 Poznań, Poland; kania.karolina@spsk2.pl; 3Division of Neurochemistry and Neuropathology, Department of Neurology, Poznan University of Medical Sciences (PUMS), 60-355 Poznań, Poland

**Keywords:** multiple sclerosis (MS), immunology, autoimmunity, vaccination, disease modifying therapy (DMT), vaccination immunology, safety

## Abstract

Vaccines work by stimulating the immune system, and their immunogenicity is key in achieving protection against specific pathogens. Questions have been raised whether in Multiple Sclerosis (MS) patients they could induce disease exacerbation and whether vaccines could possibly act as a trigger in the onset of MS in susceptible populations. So far, no correlation has been found between the vaccinations against influenza, hepatitis B, tetanus, human papillomavirus, measles, mumps, rubella, varicella zoster, tuberculosis, yellow fever, or typhoid fever and the risk of MS. Further research is needed for the potential protective implications of the tetanus and Bacillus Calmette–Guerin vaccines in MS patients. Nowadays with the emerging coronavirus disease 2019 (COVID-19) and recent vaccinations approval and arrival, the risk-benefit in MS patients with regards to safety and efficacy of COVID-19 vaccination in those treated with immunosuppressive therapies is of paramount importance. In this manuscript, we demonstrate how different vaccine types could be related to the immunopathogenesis of MS and discuss the risks and benefits of different vaccinations in MS patients.

## 1. Introduction

World Health Organization (WHO) estimates that between the years 2010 and 2015 more than 10 million deaths were prevented owing to vaccinations carried out around the world, making vaccines one of the most important triumph stories of modern age medicine [1]. On the other hand, vaccines have been a source of public controversy with regards to their safety. While generally considered safe in people with healthy immune systems, a special consideration needs to be taken when it comes to patients with altered immune status, namely with autoimmune diseases or under immunosuppression. Since vaccinations work on activating the immune system, it has been hypothesized that a stimulus of the immune system (e.g., a vaccine) may trigger an autoimmune disease or its exacerbation [2].

Multiple sclerosis (MS) is the most common cause of nontraumatic disability in young adults worldwide [3]. There are currently over 2 million people living with MS around the world, with nearly 1 million in the US alone [3,4]. MS is a condition of the central nervous system (CNS) with proven autoimmune pathology. While the disease is incurable and, thus far, the scientific community has not been able to induce tolerance towards myelin self-antigens, MS is controlled by immunomodulating or immunosuppressive therapies.

Both autoimmunity and immune therapies are potentially problematic with regards to vaccinating MS patients. Hence, many patients with MS and their physicians face an ongoing dilemma on whether or not to vaccinate. This should be especially important nowadays, when we are facing a global pandemic caused by severe acute respiratory syndrome coronavirus 2 (SARS-CoV-2) which causes coronavirus disease 2019 (COVID-19), with vaccinations against SARS-CoV-2 approaching. Several aspects regarding MS must be taken into consideration when discussing immunizations. The aim of this review is to demonstrate how different vaccine types could be related to the immunopathogenesis of MS and to discuss the risks and benefits of different vaccinations in MS patients.

### 1.1. Immunopathophysiology of Multiple Sclerosis

MS is a chronic disease primarily driven by immune-mediated mechanisms. It is currently believed that autoreactive CNS-directed B and T cells are activated in the periphery, gaining access to the CNS and thus becoming effector cells. The mechanism of this activation is diverse and includes molecular mimicry or recognition of a CNS antigen released into the periphery from damaged CNS cells. It needs to be underscored that the primary antigen that triggers this response is yet unknown and possibly diverse in different patients [5]. Following damage to CNS tissue, resident immune cells within the CNS, particularly microglial cells, are activated. By upregulating their major histocompatibility complex (MHC) class I and II and cell surface co-stimulatory molecules, as well as secreting cytokines and chemokines, other immune cells such as CD4+ and CD8+ T cells, B cells, monocytes, macrophages and dendritic-like cells can easily find their way into CNS lesions [6]. At the same time, CNS antigens never seen by the immune system from the lesion are getting exposed and processed to be introduced to incoming T cells. CD8+ T cells recognize short peptides in the context of MHC class I, while CD4+ T cells recognize these peptides in the context of MHC class II molecules [7]. Major pathophysiological mechanisms of MS involve autoreactive Th-17 cells and T helper Th-1 CD4+ T cells which secrete interleukin IL-17 and IL-22 and interferon gamma (IFN-ү), respectively [8]. Th-17 cells increase the secretion of proinflammatory molecules, activate microglia, recruit other inflammatory cells and aid in augmenting permeability of the blood brain barrier (BBB) via disrupting tight junctions on BBB endothelial cells mainly through the action of IL-17 and IL-22. Th-1 cells increase the expression of MHC molecules of cells in the CNS, thereby participating in direct killing of oligodendrocytes and activating microglia [8,9].

Each cell has its own role in the pathogenesis of MS. While CD4+ T cells recruit macrophages, the later release proinflammatory cytokines and toxic molecules; CD8+ T cells can directly attack MHC class I-expressing cells such as oligodendrocytes and neurons, and finally, B cells are stimulated and produce the pathogenic autoantibodies that aid myelin destruction [7]. The intrathecal production of humoral immune response, namely oligoclonal bands, represents an inseparable part of the disease [10]. However, the exact target specificity of these autoantibodies remains unknown and seems patient-specific [11].

Various autoreactive immunoglobulins found in MS patients’ cerebral spinal fluid (CSF) and serum have been acknowledged to target CNS antigens and further support the involvement of humoral immunity in the pathogenesis of MS [12]. Of note are the intrathecal antibodies against Measles, Rubella and Varicella Zoster viruses [13] (see the discussion section). Another factor holding diagnostic and prognostic values include the light subunit of neurofilaments (NFL), which correlate with axonal injury and displays higher prevalence in progressive MS [12].

The ultimate effects of this inflammatory cascade, myelin sheath damage and BBB breakdown, lead to further axonal damage, loss and dysfunction as well as demyelination of CNS, which contribute to the clinical presentation of MS patients [6].

### 1.2. Multiple Sclerosis Epidemiology

MS epidemiology is broad spectrum, comprising of genetic, environmental and infectious causes. Clinical studies support that genetic factors are directly linked to an increased risk of developing MS although there is no evidence that MS is directly inherited. A study done in Canada demonstrated an increased familial risk of disease as high as 300-fold for monozygotic twins and up to 40-fold for biological 1st-degree relatives of patients with MS [14]. Another study which was done in Canada as well, focused on conjugal MS couples (both parents have MS) and assessed the recurrence risk in progeny of such pairs. The study estimated that the offspring of such relationships have a considerably increased risk of developing MS compared to both the lifetime risk for MS in the general population in Canada, which is approximately 0.2%, and the crude risk for MS in children of matings with only 1 affected parent which is estimated to be 0.7% [15].

In addition, the human leukocyte antigens (HLA), HLA-DR1501 and HLA-DQ0601 alleles, which encode for restriction elements of T lymphocytes, are associated with up to 4-fold increased risk of developing MS in Caucasian populations [16]. Besides a genetic association, studies have suggested that certain environmental factors such as low vitamin D levels and smoking may make specific individuals more susceptible to the disease [17]. Infectious mononucleosis caused by Epstein Barr virus (EBV) is also believed to play a major role in increasing the risk of developing MS [18]. MS also occurs more frequently in women and certain ethnic groups, including African-Americans, Asians and Hispanics/Latinos, but is most recurrently in Caucasians of northern European ancestry [19].

### 1.3. Clinical Course of Multiple Sclerosis and Associated Pathology

There are several distinct disease courses: relapsing remitting MS (RRMS), primary progressive MS (PPMS), secondary progressive MS (SPMS), clinically isolated syndrome (CIS), and radiologically isolated syndrome (RIS) [20].

RRMS is found in about 85% of MS patients, making it the most common subtype. It is identified by alternating periods of clearly defined neurological impairment such as bouts of weakness and fatigue, anomalous sensation, and balance and vision damage (defined as relapses) and periods of partial or complete recovery in which symptoms can either disappear or become permanent; during these, a patient is considered relatively clinically stable (defined as remissions). Of note, there is no apparent progression of the disease during remission periods [20].

PPMS is characterized by ongoing, worsening progression of neurological dysfunction from symptom onset, with accumulation of incapacities and lack of relapses and remission found in RRMS [20].

SPMS is traditionally detected retrospectively, following a primary relapsing and remitting course after which untreated patients ultimately advance into SPMS. Disease burden steadily increases as clinical disability becomes more apparent and relapses become less frequent and eventually disappear [20].

CIS is defined as the first episode of neurological demyelinating CNS pathology symptoms which last more than 24 h. It does not meet the criteria for MS diagnosis due to lack of dissemination in time and space [21]. These may include optic neuritis (ON), transverse myelitis, brainstem syndromes and cerebral hemispheres pathologies. CIS may or may not progress into clinically definitive MS in the future time. The presence and amount of brain magnetic resonance imaging (MRI) abnormalities are highly suggestive of risk of progression to definite MS conversion [21,22].

Although the last is not considered a separate MS phenotype, RIS identifies patients with CNS anomalies on MRI reminiscent of MS demyelination lesions not explained by another diagnosis, in the absence of neurological symptoms on physical examination [20]. There are few clinical studies that support a strong likelihood of patients diagnosed with RIS to further develop radiological and clinical progression or even meeting the criteria for MS years after being diagnosed with RIS [23,24,25,26].

The hallmark pathology of MS is plaques. They affect chiefly white matter of the brain, spinal cord and optic nerve but may involve cerebral cortex as well. The plaques are essentially a combination of inflammation, demyelination and axonal injury or loss [27]. Although they are found in all subtypes of MS, plaques exhibit fundamental differences. These profound heterogenicities manifest as variable degrees of different cell damage levels and inflammatory response and can be histologically distinguished as active, chronic or remyelinated lesions [28]. Active lesions are common in RRMS. They are identified as distinctive focal plaques of demyelination, inconstant degree of axonal loss that is relatively preservative, perivascular and parenchymal inflammatory infiltrates and reactive gliosis [28]. Remyelinated lesions are commonly seen at the boundaries of these active plaques. They encompass thin myelinated axons and numerous oligodendrocyte precursor cells [28]. Chronic lesions, which are seen in the progressive courses, display more extensive demyelination and axonal injury and limited low-grade active inflammation with diffuse gray and white matter atrophy [27,28].

### 1.4. Immune Mechanisms of Different Types of Vaccinations

The two fundamental halves of the immune system are the innate and adaptive immunities. These cooperate with one another repeatedly throughout life, each taking care of different tasks to provide an efficacious and continuous immune response [29].

Innate immunity provides an immediate response through recognizing danger signals found on pathogens. Situated at “hot” susceptible anatomic sites, the physical barriers and internal defenses, such as acidic environments, temperature changes, mucus and cilia components, complement pathways and various cell types including mononuclear phagocytes, granulocytic cells, natural killer cells and dendritic cells and associated products like lysozymes, interferons and collectins, being equipped with various tools allowing them to quickly fight nonspecific infections. The innate immune system can either work independently and potentially eradicate pathogens without assistance from the adaptive system, or it can combine forces with and stimulate the adaptive system to become involved [29].

Adaptive immunity which is slower in response acts through recognizing specific proteins ultimately inducing tools for targeting specific pathogens. It is composed of B cells and T cells that exert their effects through humoral-immunity by means of antibodies, and cell-mediated immunity by means of CD4+ helper cells and CD8+ cytotoxic cells, respectively. It utilizes an ongoing mechanism that encourages a memory response that lasts years. This acquired immunity is achieved by either passive or active means and can be done by natural or artificial sources [29].

Vaccine-induced protection is achieved either by antibodies, T cell-dependent factors or by a combination of the two, which ultimately induces a cascade of mechanisms and mediators such as cytokines and neutralizing or antitoxic antibodies.

There are 5 main types of vaccines: live-attenuated, inactivated, toxoid, subunit\recombinant\polysaccharide\conjugate and RNA vaccines. Because their characteristics are diverse, each vaccine can stimulate the immune system in a different way, creating a unique set of advantages and drawbacks.

#### 1.4.1. Live-Attenuated Vaccines

These are composed of a weakened form of a disease-causing pathogen. Examples include measles mumps rubella (MMR), rotavirus, smallpox, chickenpox, adenovirus, typhoid Ty21a, Bacillus Calmette–Guérin (BCG) and yellow fever [30]. In order to produce a proper immune response, the pathogen must replicate in the vaccinated person [30]. By “mimicking” the natural infection, both antibody and cellular-mediated immune responses are created, and a robust and long-lasting immune response is achieved [29]. When a disease ensues, it is usually milder than the natural disease and is seen as an adverse reaction to the vaccine [30]. Since a measured amount of a live pathogen does exist in the vaccine, people with weakened immune systems are at risk of uncontrolled replication of the pathogen within their bodies and therefore should not receive these vaccines [30]. In addition, there is always a possibility of reversal to the original virulent form of the pathogen and a realistic threat to becoming infected [29].

#### 1.4.2. Inactivated Vaccines

These are composed of a killed germ produced in the lab by means of chemicals, heat or radiation [29]. Examples include hepatitis A, influenza, polio and rabies [30]. By destroying the pathogen and taking away the pathogen’s ability to replicate, both a safer and a weaker immune response is produced, so they can be given to immunocompromised patients [30]. Unlike live vaccines, inactivated vaccines mainly produce an antibody response with little to no cellular immune response [30]. Antibody titers against inactivated vaccines are wanning with time necessitating boosters to be administered over time to maintain immunity [30].

#### 1.4.3. Toxoid Vaccines

The germ produces a toxin which acts as the harmful product that causes the disease. These toxins are inactivated by formalin and are used as targets for the immune system [29]. Examples include diphtheria and tetanus. Like some other types of vaccines, booster shots are needed to maintain a sustained level of protection against these diseases [29].

#### 1.4.4. Subunit\Recombinant\Polysaccharide\Conjugate Vaccines

Instead of using the entire pathogen, specific pieces of a germ like protein, sugar or capsid, referred to as antigens that can stimulate the immune system, are selected to be used. Examples include Haemophilus influenzae B Hib, hepatitis B, human papillomavirus (HPV), whooping cough, pneumococcal disease, meningococcal disease and shingles [30]. Because these vaccines use a very specific part of the germ, a strong immune response is made, although boosters may be needed for ongoing protection [30]. Subunit vaccines are usually combined with adjuvants, which aid in modulating vaccine’s immunogenicity in order to elicit a sturdy immune response, because antigens alone are not sufficient to induce adequate long-term immunity [2]. The most commonly used adjuvant is aluminum salts; others include squalene and monophosphoryl lipid a (MPLA), oil emulsions, saponin, toll-like receptor (TLR) agonists, enterotoxins, polysaccharides, and glycolipid adjuvants [2].

#### 1.4.5. RNA Vaccines

Instead of using the virus or the antigenic protein itself, RNA vaccines deliver the genetic code of the antigen to be endogenously expressed. Compared to other vaccine types, it has no risk of reversion to virulence, can elicit both cellular and humoral immunity, avoids anti-vector immunity and can be manufactured quickly and inexpensively. These allow for rapid deployment during emergencies, as it has been done during the COVID-19 pandemic. There are two types of RNA vaccines: conventional mRNA-based vaccines, that encode only the antigen of interest, and self-amplifying mRNA vaccines, that also encode the viral replication machinery, enabling intracellular RNA amplification and abundant protein expression [31,32]. They are encapsulated in order to prevent their degradation by RNAases and to facilitate their uptake by dendritic cells. This capsule could be made of either chitosan nanoparticlas, polyethyleinimine-based polyplexes, or lipoplexes [33]. So far, RNA vaccines have been developed against COVID-19, Ebola, zika, rabies and influenza viruses, the first three already registered for medical use. This technology has been used in non-infectious diseases as well, namely as cancer immunotherapy [31].

### 1.5. The Basic Mechanisms of Disease Modifying Therapies for Multiple Sclerosis

The mechanisms of action of disease modifying therapies (DMTs) are still incompletely understood, especially in case of the older agents. Most of them ultimately result in a shift from the proinflammatory Th-1 response to the less inflammatory Th-2 response.

Interferon-beta (IFN-β) was the first DMT approved for MS, but its pleiotropic mechanism of action in MS is still uncertain. It suppresses inflammation directly by increasing the production of anti-inflammatory cytokines (IL-10) and indirectly by increasing production of CD56-bright-natural-killer cells, which are the known producers of anti-inflammatory agents [34]. Moreover, it inhibits proinflammatory cytokines, limits leukocyte migration through the BBB and increases the production of nerve growth factor, which may lead to potential neuronal repair [34].

Another platform injectable therapy, namely glatiramer acetate (GA), which is a copolymer of four amino acids resembling myelin basic protein, acts as a T cell receptor antagonist. It also increases the secretion of anti-inflammatory agents by leading to preferential differentiation of CD4+ T cells into T helper cells [35]. Other mechanisms of action include the immunomodulatory effect on B cells and stimulation of oligodendrogenesis by activation of oligodendrocyte precursor cells [36].

An oral formulation, dimethyl fumarate, decreases absolute lymphocyte counts by inducing T cells apoptosis, but the clear mode of action remains uncertain. It particularly reduces the pro-inflammatory subsets (Th-1 and Th-12) as well as circulating memory B cells [37,38].

Another first-line oral drug, namely teriflunomide, inhibits dihydro-orotate dehydrogenase, a mitochondrial enzyme, which reduces proliferation of T and B lymphocytes by interfering in their pyrimidine synthesis pathway [39].

Both platform therapies and the above-mentioned oral drugs have a moderate efficacy in decreasing relapse rate (compared to placebo, they decrease annual relapse rate from 30% for injectables and teriflunomide to 45% for dimethyl fumarate) and radiological activity of the disease. Their efficacy in decreasing disability in the long-term is still debatable.

Highly effective therapies usually have more direct and specific mechanisms of action. They have better odds of controlling relapses and radiological activity of the disease, and their long-term effects on disability are more established. However, they are associated with higher risks for patients, including (but not limited to) opportunistic infections.

Among them are three anti-monoclonal antibodies, natalizumab, alemtuzumab and ocrelizumab, and the oral agents, namely, sphingosine 1-phosphate receptor (S1PR) modulators and cladribine.

The highly potent Natalizumab binds α4-integrin found on leukocytes and disables their migration across the BBB, as α4-integrin interacts with vascular-cell adhesion molecule 1 (VCAM-1) on brain cells [40]. Natalizumab had a nearly 70% efficacy in decreasing relapse rate compared to placebo in its pivotal clinical trial [41]. It also has a fast and robust effect on radiological activity of the disease. It belongs to the so-called maintenance therapies, which need to be administered regularly (in this case every month) to be effective. Moreover, when withdrawn (for instance for the risk of progressive multifocal encephalopathy, which is the most dangerous of its potential side effects), it can lead to a severe rebound of disease activity.

Alemtuzumab is a humanized monoclonal antibody against CD52, which leads to prolonged T cell lymphopenia and transient B cell lymphopenia [40]. It is a prototypical immune reconstitution therapy, which causes long-term effects after two yearly courses of infusions. It is designed to cause a depletion phase, followed by repopulation. Although highly effective (49–55% of annual relapse rate reduction compared to an active drug, namely IFN-β-1A three times a week), it can potentially generate autoimmune complications (especially thyroid-related) and is associated with an increased cardiovascular risk (such as hemorrhagic stroke, myocardial infarction and artery dissection) [42].

Ocrelizumab acts by binding to CD20 which leads to selective B cell depletion via apoptosis, antibody-dependent cellular cytotoxicity and cell-mediated phagocytosis [43]. It needs to be administered intravenously every 6 months in order to modify the disease. In clinical trials, it was shown to reduce relapse activity by nearly 50%, compared to IFN-β-1A three times a week, and practically eliminate radiological activity of the disease [44]. The potential risks are those associated with lymphopenia (infections), reactivation of hepatitis B if diagnosed in a patient, and infusion-associated reactions which tend to diminish with therapy duration.

The other group of DMTs are S1PR modulators. They antagonize the receptor’s function leading to lymphopenia by sequestration of lymphocytes in the lymph nodes [45]. The first modulator to be registered was fingolimod (53% efficacy of relapse rate reduction compared to placebo) [46] and the subsequent are siponimod (approved in SPMS), ozanimod (approved in 2020) and ponesimod (approved by FDA in 2021) [45,47]. These are also maintenance therapies, which could be associated with rebound MS activity when withdrawn, as documented for fingolimod. Their risks include, among others, opportunistic infections, clinically significant bradycardia and macular edema [46].

Cladribine, a purine nucleoside analogue, is considered a selective immune reconstitution therapy. Its active metabolite accumulates preferentially in lymphocytes resulting in their DNA damage and apoptosis. Ultimately, it causes temporary lymphocytes depletion followed by gradual reconstitution of peripheral counts over time [48]. This is another example of immune reconstitution therapy, which is designed to cause a long-term qualitative change in the immune system. In clinical trials, it decreased relapse rate by nearly 60% compared to placebo and was associated with increased risk of Herpes or Varicella-Zoster infections [49].

As demonstrated, all of the above-mentioned mechanisms of action could potentially interfere with vaccination response, especially if lymphocyte depletion or anergia are present. Inadequate response to vaccination, and not increased risk of complications, is a major concern here. However, immunosuppression could potentially increase the risk associated with live attenuated vaccines.

## 2. Vaccination Overview in the Context of Multiple Sclerosis

### 2.1. Influenza

Influenza is an enveloped negative-strand RNA virus from the Orthomyxoviridae family. There are three types based on genetic and antigenic differences: A, B, and C. Type A, the most clinically significant, is the only one holding a significant risk of zoonotic infection and host switch, posing a potential threat of emerging influenza pandemics. Challenges with eradication of the virus stem from its ability to infect many animal species and having a natural reservoir in birds.

Influenza viruses mutate rapidly, in part due to the lack of proof-reading activity in their polymerase complex, leading to alterations in all proteins, including hyaluronidase (HA) and neuraminidase (NA) epitopes against which antibodies are made. This gradual change is called antigenic drift and is responsible for the need of new yearly vaccines against influenza [50,51].

A study on rats explored the increased risk of exacerbation in MS patients following an upper respiratory tract infection (URI). It demonstrated that the lungs could act as a niche for T lymphocytes, including myelin-reactive memory T cells that have the ability to travel to the CNS after lung infection or inflammation. The study also found that during an experimental autoimmune encephalomyelitis (EAE) episode in mice, virus-specific CD8+ T cell responses were diminished, possibly explaining the increased morbidity of MS patients during infection. It is important to note that the animals were infected with influenza virus and exacerbations of EAE were not necessarily duplicated with other lung inflaming organisms. Even though URIs are most common in winter and autumn seasons, an analysis showed that MS exacerbations are in fact less frequent during this time period. This contradictory evidence is possibly explained by melatonin variability and its ability to drive T cell differentiation and affecting autoimmune activity [52].

The first vaccination attempt with a live influenza vaccine occurred in 1936 in the Soviet Union. This type of vaccine faced many challenges and is currently used only in immunocompetent children aged 2–7 years. Due to the complicated usage of live-attenuated vaccines, programs attempted to develop an inactivated form of the vaccine. Those attempts were greatly intensified at the beginning of WWII with the objective of protecting troops. In 1942, the first doses of a bivalent inactivated vaccine against influenza A and B were administered and proved successful with a 69% efficacy rate in an epidemic just over a year later [53].

Currently, there are two main types of influenza vaccines: inactivated injectable, a killed vaccine administered intramuscularly, and live-attenuated, administered intranasally by using an aerosol spray [54]. Multiple studies regarding two prevalent inactivated vaccines (FOCETRIA—seasonal [55]; PANDEMRIX—pandemic [56]) revealed two common findings: (1) An increased risk of acute hospitalization that correlates with exacerbations among MS patients was noted with the onset around the time of clinical influenza infection. (2) No additional risk of acute hospitalizations following neither the pandemic or seasonal influenza vaccinations was identified among MS patients [57,58].

The authors could not find any studies exploring the relationship between live-attenuated vaccines and MS patients. Nonetheless, several case reports demonstrating demyelinating processes in nature that occurred following live-attenuated or inactivated vaccine administrations were reported and are worth mentioning. These included longitudinally extensive transverse myelitis [59], bilateral ON [60] and acute disseminated encephalomyelitis (ADEM) [61] in healthy individuals. The transverse myelitis case was reported in a 27-year-old female who developed symptoms 4 days following a nasal attenuated novel influenza A (H1N1) vaccination [52]. This was in a setting of 2009 widespread vaccinations against the novel strain. The patient underwent extensive diagnostic work-up. Of note, oligoclonal bands and anti-aquaporin 4 antibodies were absent. Her CSF exam revealed a marked lymphocytic pleocytosis of 517 cells/μL. She improved significantly with plasma exchange and intravenous steroids with a 6-week oral taper. Her brain MRI was unrevealing. The ON case was in a 23-year-old female who received an inactivated influenza vaccine 2 weeks prior to symptoms onset [53]. On MRI, she had enhancement of the right optic nerve and leptomeninges of the right frontoparietal lobes. Following the initial involvement of the right optic nerve, which improved without treatment, left-sided symptoms occurred 3 weeks later. Oral steroids were used but given the following recurrence of the symptoms in the right eye, intravenous methylprednisolone and plasma exchange were used. As in the previous case, oligoclonal bands and anti-aquaporin 4 antibodies were negative. The patient even underwent leptomeningeal and brain biopsy, which showed reactive astrogliosis and lymphocyte infiltration. Both cases improved, and both were highly atypical for multiple sclerosis.

The influenza vaccine is the most widely studied vaccine in association with MS. Following vaccination, the production of adequate immune response and achievement of sufficient seroprotection significantly change from one therapeutic agent to another.

Those studies assess the effects of different therapy options for MS and vaccine efficacy since some have demonstrated to interfere with immunogenicity mechanisms in building a response for this vaccine. Most drugs do not appear to reduce the response to the influenza vaccine.

Patients receiving IFN-β had equivalent efficacy for the influenza vaccine and showed decent antibody protection levels compared to both healthy controls and untreated MS patients [62,63].

Patients receiving GA were found to have a lower grade of protection following influenza vaccine compared to healthy individuals, although the lack of vast clinical data on this drug regarding this vaccine warrants further studies to be done [62].

Patients receiving natalizumab face a conflicting situation. There are 2 clinical studies that demonstrate 2 different outcomes: one reporting a lower rate of protection [62] and the other reporting no significant differences between MS patients and healthy controls [64]. It is probable that relatively small sample sizes used in these trials can potentially account for these contradictory outcomes, again, necessitating further analyses to resolve the topic. The mode of action of natalizumab does not suggest interference with vaccination response.

Patients receiving fingolimod were able to mount a proper immune response following the influenza vaccine compared to healthy individuals; nevertheless, seroprotection rates steadily declined at 3 and 6 weeks post-vaccination when compared to placebo-treated patients [65]. This reduced response rates should be carefully kept in mind when vaccinating patients on fingolimod therapy for the potential need for booster shots.

No studies were found regarding dimethyl fumarate and influenza vaccine. However, the immune responses to the inactivated vaccines, tetanus-diphtheria toxoid (Td), pneumococcal polyvalent vaccine and meningococcal (groups A, C, W-135, and Y) oligosaccharide, were found adequate, and seroprotection levels were achieved in MS patients treated with delayed-release dimethyl fumarate in comparison to patients receiving non-pegylated IFN [66].

Patients receiving mitoxantrone were found to have no protection after receiving the influenza vaccine, which indicates that mitoxantrone impairs influenza vaccine immunogenicity and efficacy [62].

Patients receiving teriflunomide were able to mount a slightly reduced response to the vaccine when compared to patients treated with IFN-β; however, it was still considered sufficient to be considered protective [67].

Although there are no studies up to date that explore the relationship between patients taking alemtuzumab and the influenza vaccine, a small pilot case-control study did assess the connection between three other T cell dependent and independent vaccines (tetanus diphtheria and polio, meningococcus C and pneumococcus). They concluded that patients treated with alemtuzumab were able to mount a normal humoral immune response [68]. Still, the timing of vaccination and alemtuzumab treatment phase are of utmost importance—should vaccination be administered during the depletion phase, patients potentially might not be able to mount a successful response.

Of similar note, no studies were found in patients taking cladribine, though a recent fatal case of influenza A associated pneumonia within months of administering subcutaneous cladribine causing critical severe lymphopenia was reported [69]. This raises a question of how to carefully plan MS treatments in the context of not only the disease itself but also of the risk associated with seasonal flu infections and the necessity to administer appropriate vaccinations before the cladribine course is started.

Patients receiving ocrelizumab, who were studied in the VELOCE study, did mount a response to the influenza vaccine, although it was attenuated compared to the control groups (receiving either IFN-β or no DMT) [70].

For MS patients, type and timing of vaccine (with inactivated influenza virus or live attenuated) combined with MS specific medications (IFN-β versus non-IFN-β medications) should be considered carefully. Influenza infections may cause worsening of MS-related symptoms and trigger new relapses and should be prevented when possible. There is no sufficient proof that influenza vaccination could induce MS exacerbation.

### 2.2. Hepatitis B Virus (HBV)

HBV is a partially double-stranded DNA virus from the Hepadnaviridae family. It is the cause of Hepatitis B, the most common chronic viral infection. Due to its possible asymptomatic presentation on acute infection, many people are unaware of their infection and can become chronic carriers of the virus. The chronic form is mostly asymptomatic as well but may progress to liver cirrhosis and hepatocellular carcinoma. According to the WHO, more than 2 billion people in the world have positive serologic markers for hepatitis B infection, including more than 350 million chronic carriers [71,72,73].

A food and drug administration (FDA) approved plasma-derived HBV vaccine became available in 1982. However, this vaccine was later withdrawn in 1986, when HBV antigens were successfully made in yeast. It is the first vaccine against major human cancer and is part of present routine immunization programs in many countries. Currently, all recombinant vaccines containing hepatitis B surface antigen (HBsAg) are expressed in yeast [74].

After a mass immunization campaign in France between 1995–1997, several cases of MS were reported a few weeks after HBV vaccination. This led to a temporary suspension of the school-based hepatitis B vaccination program in 1998 and generated widespread concern in other countries. The decision was based on two case-control studies conducted in the UK and France, with both showing a nonsignificant increase in risk of developing MS following the HBV vaccine [75].

Another case-control study nested within the General Practice Research Database (GPRD), which includes over 3 million Britons, found that immunization against hepatitis B was associated with a threefold increase in the incidence of MS within 3 years following the vaccination. It is important to note, however, that 93% of MS cases in the study were not vaccinated. Moreover, the practice of hepatitis B vaccination in the UK at the time of the study was targeted toward high-risk individuals, which might have biased the results [76,77].

An ecological study in Canada comparing the incidences of MS and post-infectious encephalomyelitis between adolescents in the pre-vaccination (prior to 1992) and post-vaccination periods showed no evidence for a relation between hepatitis B vaccination at age 11–12 years and the subsequent onset of MS or post-infectious encephalomyelitis [78].

A nested case-control study in two large cohorts of nurses in the US using both healthy women and women with breast cancer as controls (to address the potential recall bias among women with a serious disease) demonstrated no association between hepatitis B vaccine and risk of MS in women [75].

A retrospective case-control study that included both men and women in 3 large health maintenance organizations (HMOs) evaluated the risk of CNS demyelinating diseases in adults due to several vaccinations. The study found that hepatitis B vaccine and several other vaccines were not associated with an increased risk of MS or ON [79].

Some MS drugs have been associated with the reactivation of HBV in patients. One such drug is ocrelizumab. Prior to initiation of ocrelizumab treatment, patients should be screened for HBV. The drug is contraindicated in persons with active HBV infection [80].

HBV status should also be checked prior to initiating ofatumumab, a human monoclonal antibody to CD20. If patients are found to be HBsAg positive, they should receive potent prophylactic oral antivirals against hepatitis B [81].

Another example is alemtuzumab. It is important to check for serological markers for HBV prior to initiating treatment. Some patients may still carry the virus while being HBsAg negative (occult hepatitis B), which can be detected using more sensitive modalities, like polymerase chain reaction (PCR) [82]. An additional example is the purine analog, cladribine [83,84].

Based on the evidence of the improbable association of HBV vaccine and the risk of MS, and the possibility of HBV reactivation as a side effect of some of the MS drugs, it is recommended to follow the current HBV vaccination guidelines.

### 2.3. Tetanus

Tetanus is a life-threatening disease caused by an infection with Clostridium tetani, a spore forming bacterium commonly found in soil and gastrointestinal tract of humans and animals. It is caused by tetanospasmin, a potent neurotoxin, and manifests clinically with painful spasms and muscular rigidity [85]. The disease is preventable with the tetanus vaccine, which is also used in some cases for post-exposure prophylaxis. Also known as tetanus toxoid (TT), it is an inactivated toxoid vaccine that is given together with the vaccines against Diphtheria and/or Pertussis in the forms of DTaP, Tdap, DT, and Td. Due to immunity decreasing with time, which is typical with this vaccine, booster doses are recommended by the center for disease control (CDC) every 10 years for adults.

A systematic review on vaccinations and MS risk followed eight studies that assessed the risk of developing MS after tetanus vaccination [86]. None showed an increased risk of developing MS, while three demonstrated a potential protective effect of the vaccine from developing MS [79,87,88]. An additional systematic review study has demonstrated a reduced risk of MS in people who were vaccinated for tetanus [89]. Another study has demonstrated that immunity to tetanus, either naturally acquired or vaccine induced, is associated with a decreased risk of developing and the progression of MS [90]. Confavreux et al. found a decreased risk of relapse of MS following tetanus vaccination, but it was not statistically significant [91].

Based on the evidence presented by multiple studies demonstrating no increased risk of MS development or relapse with tetanus vaccination, with some studies actually showing decreased risk, the tetanus vaccine should be recommended to MS patients when indicated.

### 2.4. Human Papillomavirus (HPV)

HPV is the most common sexually transmitted disease worldwide [92]. It is associated with various dermatological conditions such as cutaneous and anogenital warts and neoplastic illnesses such as cervical, vulvar and vaginal cancer in women, penile cancer in men and anal and mucosal cancers in both genders [93].

Most HPV infections clear out by themselves within the first 2 years of infection thanks to rapid immune response. Though there are hundreds of HPV types, less than a dozen are referred to as high-risk and carcinogenic. These HPV infections usually last longer and can eventually progress to cancer. Of these, the two types HPV-16 and HPV-18 are the commonest ones to be associated with cervical cancers [92].

There are currently 3 types of HPV vaccines: quadrivalent, bivalent and nonavalent vaccines, estimated to prevent up to 90% of cancers caused by HPV from ever developing [92].

Based on a systematic review published in 2018 that included various observational studies, reviews and a randomized clinical trial pooled analysis, there was no association between the quadrivalent vaccine GARDASIL and the risk for MS or other central demyelinating diseases [94].

On the other hand, associations have been made between patients taking fingolimod and chronic, treatment refractory warts and even HPV-associated malignancies [95,96]. Improvement of said conditions was noticed with dose reduction or cessation of fingolimod [95]. The mechanism was thought to be a combo of impaired immune response and decreased cancer surveillance secondary to lymphocyte sequestration properties of fingolimod [95]. A thorough history of HPV infections and vaccinations as well as gynecological exams and Pap smears should be considered by physicians prior to starting fingolimod. Increased frequency of surveillance should also be encouraged for women with positive high-risk HPV strains or dysplastic changes secondary to HPV infection who are taking fingolimod. As smoking can increase the risk of perineal and oromucosal malignancies and further aggravate MS, concurrent smoking should be strongly discontinued.

### 2.5. Measles, Mumps and Rubella (MMR)

The MMR vaccine, a highly efficacious live-attenuated vaccine against the three preventable childhood viral illnesses, was first introduced in the US in 1971 [97]. This vaccine is included in the routine immunization schedules of children nowadays and given in two doses. It is also given to women who wish to become pregnant, health care workers, military staff and travelers to certain countries [98].

Measles virus, a member of the Paramyxoviridae family, is spread by inhalation of respiratory droplets. It causes fever, maculopapular rash, cough, coryza and conjunctivitis, as well as complications like pneumonia, diarrhea and the fatal subacute sclerosing panencephalitis (SSPE) that may occur 7–10 years following an acute infection [97].

Mumps virus, also a member of the Paramyxoviridae family, is spread by inhalation of respiratory droplets. It causes fever, swelling and tenderness of salivary glands (most frequently the parotid gland), orchitis, encephalitis and meningitis [97].

Rubella virus, a member of the Togaviridae family, is spread from an infected person to another by sneezing or coughing. Postnatal rubella causes a generalized maculopapular rash, fever, arthritis, lymphadenopathy and conjunctivitis. Congenital rubella syndrome, acquired in utero, causes cataracts and a unique set of birth defects including cardiac abnormalities and deafness [97].

It has been argued more than once that childhood viral infections can potentially be involved in MS pathogenesis. Indeed, childhood history of rubella and measles infections increase the risk of acquiring MS significantly as high as 1.7 times [99].

Of special interest, there were several cases of ON following days to weeks of the MMR vaccine administration [100]. All except one case, in a 13-year-old patient [101], showed significant improvement following treatment and did not develop clinically definite MS. The 13-year-old patient was diagnosed with unilateral ON and subsequent MS four weeks after rubella vaccination. Besides ON, cases of transverse myelitis following the MMR vaccine were also reported [102,103]. To the best of the authors’ knowledge, observation periods following these events were not mentioned in the above cases. Since MS can still develop up to 10 years following an acute demyelinating attack, the cases presented may be potentially of limited use.

However, several studies that investigated the relationship between the risk of acquiring MS following the measles, mumps, and/or rubella vaccines were negative. All studies but one [104] found no association between developing MS and the MMR vaccination [79,105,106,107]. The singular study that found an increased risk of MS was in fact limited due to a small number of participants that ultimately brought upon large confidence intervals of the odds ratio [104].

In accordance with immunosuppressive therapies guidelines, since the MMR vaccine is a live-attenuated vaccine, its administration should be avoided in MS patients receiving these treatments.

### 2.6. Varicella-Zoster Virus (VZV)

VZV is a ubiquitous, highly neurotropic herpes virus. Primary infection causes varicella (chickenpox); this disease is highly common during childhood which is usually harmless in healthy children whose immune system controls the infection. Later, VZV establishes latency in ganglionic neurons, and its reactivation causes zoster (shingles) [108].

Following initial exposure, the innate immune system aids in shifting the local mucosal site defense lines to a global and long-lasting VZV-specific immunity. While IgG antibodies neutralize the virus, T cells are of paramount importance because they prevent symptomatic disease after both re-exposures and reactivations of endogenous VZV. Hence, treatments targeting T cells have the potential to diminish the immune response to VZV and increase the risk of Herpes Zoster (HZ) [109].

Importantly, most DMTs in MS affect T cell mediated immunity. A special awareness must be taken for patients taking fingolimod [110]. There are numerous clinical trials that support an increased incidence of HZ infections in fingolimod treated patients [111,112]. Although serious or complicated cases of HZ were uncommon, several fatal cases were reported and include VZV encephalitis, VZV vasculitis [113] and primary varicella zoster infection [114].

Other DMTs linked to VZV infections are natalizumab [115], associated with VZV meningitis, meningoencephalitis and meningomyelitis [116]; and alemtuzumab, associated with VZV meningitis and multidermatomal VZ [117].

Currently there are two types of shingles vaccines in the US that are recommended for adults aged 60 and above, regardless of previous infection: live-attenuated (Zostavax; licensed at 2006, given as a single subcutaneous injection) [118] and non-live recombinant (Shingrix; licensed at 2017, given as intramuscular injections, 2 doses separated by 2–6 months apart) [119]. The Advisory committee on immunization practices (ACIP) currently recommends the Shingrix as the preferred vaccine for shingles prevention.

In two clinical trials conducted on adults with autoimmune diseases, the immunogenicity and safety of the Shingrix vaccine were studied. In one study, there were 21 MS patients (out of 190 participants) [120]. The Shingrix vaccine was well tolerated and had a clinically acceptable safety profile. In addition, it elicited statistically significant VZV-specific immune responses [121].

Current guidelines recommend that the varicella vaccine, a live-attenuated vaccine, should be considered for patients with MS who never had chicken pox, are seronegative and lack prior immunity and are considering starting immunosuppressive drugs. The CDC recommends two doses four weeks apart, and patients should not start fingolimod until at least one month after the last dose of the varicella vaccine [122].

It is important to test VZV status prior to such treatments, particularly in those with an occupation that may increase the risk of exposure to the virus, including jobs in health care or schools. Screening for past exposure and antibody titers against VZV are recommended prior to initiation of treatment with immunosuppressive agents [123].

### 2.7. Tuberculosis and the Bacillus Calmette Guerin (BCG) Vaccine

Tuberculosis (TB) is an infectious disease caused by Mycobacterium tuberculosis. This airborne transmitted disease yields approximately 10 million new cases annually and is responsible for 1.6 million deaths diagnosed every year [124]. Currently, the BCG is the only available vaccine.

Based on a systematic review that included 6 different studies, administration of the BCG vaccine was not found to be associated with an increased risk of neither developing MS nor increased relapse rates in MS patients [125].

In fact, it has been demonstrated that following inoculation of the bacillus in BCG vaccination in both the animal model of MS (EAE) and human MS patients, clear beneficial effects were found. It was suggested that MS activity was reduced by altering Th-17 responses and affecting IFN-ү levels, both being classical proinflammatory factors in MS [124].

A randomized study on 73 patients diagnosed with CIS revealed beneficial effects of the BCG vaccine in the short and long-term. It significantly decreased disease activity within the first 6 months of administration as it decreased both the number of T1-hypointense lesions and collective number of relapses following 18 months. It was also found to reduce the risk of developing MS over a period of 5 years [126].

Since many DMTs affect the immune system, they can potentially facilitate reactivation of TB [127]. This risk was not reported in MS patients taking IFN-β or GA, making them the safest treatments for MS patients with latent TB [127]. In contrast, the drugs used in treatment of MS that were associated with cases of TB reactivation are fingolimod, natalizumab, glucocorticosteroids, azathioprine, cyclophosphamide, methotrexate, alemtuzumab, ocrelizumab and cladribine. Therefore, recommendations for these patients vary from pre-treatment screening, BCG vaccinations and even strict avoidance of these drugs [127,128]. Although variances exist amongst different MS drugs, TB screening is recommended for all MS patients commencing highly immunosuppressive therapies, including ocrelizumab, alemtuzumab and cladribine [129,130].

### 2.8. Yellow Fever (YF)

YF is a vector-borne disease transmitted by a bite of an infected mosquito with the yellow fever virus (YFV). It is a potentially fatal disease endemic to sub-Saharan Africa and tropical South America, affecting approximately 200,000 people and causing 30,000 deaths annually, producing hemorrhagic fever that is fatal in 20–50% of cases. As no treatment exists for YF disease, prevention is key [131]. The YF vaccine is a live-attenuated vaccine recommended for people 9 months and older traveling to or living in high-risk areas [132].

A small study done in 2007–2009 on MS patients showed a significant increase in exacerbation rates (5 out of 7 patients experienced several relapses) within 3 months following yellow fever YD 17D vaccination compared with pre-vaccination exacerbation rates. Subsequent clinical, immunological and radiological MS exacerbations were observed [133].

In comparison, another small study done in 2014–2018 on MS patients in Switzerland showed no association with MS relapses (4 out of 23 patients developed relapses) following YF vaccinations. In addition, as previously discussed, although taking natalizumab and co-administering live-attenuated vaccines is not recommended, none of the patients experienced relapses after YFV or a YFV-related adverse event [134].

In 2018, a female MS patient demonstrated recurrent severe MS relapses following YF vaccination within two months of fingolimod withdrawal. Hence, it was indicated that in the setting of fingolimod withdrawal MS can be potentially triggered by live-attenuated vaccines [135].

These findings challenge the earlier study done in 2009 by Farez and Correale [133]. Suggested reasons include that patients were taking DMTs at the time of vaccination and the treatments differed in their efficacy. Nevertheless, both studies’ sample sizes are limited, warranting further studies to confirm these findings.

Therefore, patients with MS intending to travel to endemic YF areas must cautiously consider the potential risk of exacerbation associated with the YF vaccine against the possibility of exposure to the YFV [133].

### 2.9. Typhoid Fever (TF)

Typhoid fever is an infectious disease caused by Salmonella Typhi. It is most common in low-resource communities with poor sanitary conditions. The infection is systemic and without effective antimicrobial therapy can lead to intestinal perforation and death. Unfortunately, antimicrobial resistance of Salmonella Typhi has begun to spread and focus has shifted towards the importance of typhoid vaccines [136].

There are three types of typhoid vaccine available, Vi polysaccharide (Vi-PS), live-attenuated oral vaccine (Ty21a) and a recently approved typhoid conjugate vaccine which uses the tetanus toxoid as a carrier protein. The later became the most recommended vaccine due to its suitability for infants and young children, enhanced immunological properties and longer predicted duration of immunity [137].

Case-control studies have reached mixed results. Two studies demonstrated no significant differences between the cases and controls [138,139], an additional one demonstrated that MS patients were less likely to be vaccinated for typhoid [87], while another showed that MS patients were more likely to be vaccinated for typhoid, the latter however, was not statistically significant [88,125].

A systematic review and meta-analysis study focusing on the four case control studies mentioned has concluded that there is no association between MS and typhoid fever vaccination with an odds ratio of 1.05 and no statistical significance [125].

### 2.10. COVID-19

Emerging serious neurological complications of both the central and peripheral nervous systems following SARS-CoV-2 infection have gained more attention recently. Through invasion of the nervous system in a direct manner or by inducing a massive immune inflammatory response, namely the “cytokine storm”, neurological manifestations such as encephalopathy, stroke, encephalitis, meningitis, Guillain–Barré syndrome (GBS) and MS can result post-infection [140]. A systematic review evaluated nearly 300 COVID-19 patients found that 91% of patients who experienced some kind of a neurological complaint presented with CNS symptoms with headaches and dizziness being the commonest ones [140]. The most common CNS complication of COVID-19 was encephalopathy, which developed in about 50% of hospitalized patients mainly due to hypoxia or systemic disease [141].

In May 2020, the first case of MS following COVID-19 infection was detected. A 29-year-old woman complained of retro-orbital pain, decreased visual acuity, temporary weakness and myalgia of her limbs [142]. Physical examination discovered hyperreflexia and upper motor neuron signs as well. MRI demonstrated a right optic nerve enhancement lesion and demyelinating lesions adjacent to the lateral ventricles [142]. Further examination of her CSF done by a lumbar puncture detected the presence of IgG oligoclonal bands, but no aquaporin-4, MOG antibodies or SARS-CoV-2 were identified [142]. Following high-dose steroids her vision recovered gradually [142]. The authors underscored that at the time of ON, the patient had non-enhancing periventricular lesions suggestive of demyelination present already prior to COVID-19 infection. They suggested that the infection with SARS-CoV-2 might have been a precipitating factor of a clinically silent demyelinating condition.

Of note, several other instances of COVID-19-associated ADEM [143,144,145] and myelitis [146,147] were reported as well. Patients demonstrated variable neurological dysfunction ranging from neck and back pain, encephalopathy, limb numbness, motor and sensory weakness, visual abnormalities and urinary retention. Imaging studies were highly suggestive of acute demyelinated lesions and included diffused FLAIR hyperintense and increased T1 and T2 enhancement signals in juxtacortical, periventricular, spinal and optic areas. Some cases demonstrated oligoclonal bands or upregulated proteins levels in CSF analysis; however, others showed negative CSF findings. Treatments such as IVIG, steroids and plasma exchange brought clinical and radiological improvement and variating degrees of symptom relief [141,145,148,149,150,151].

In general, it seems that patients with MS treated with DMTs do not have a greater risk of coming down with COVID-19 or having a more severe course of infection [152]. However, several studies have suggested that anti-CD 20 therapies could increase the odds of developing COVID-19 [153,154] or severe COVID-19 [155]. In the population of MS patients, independent risk factors of severe COVID-19 were older age, obesity, higher disability assessed in expanded disability status scale (EDSS) and comorbidities [156]. However, patients with advanced disease and high disability could potentially be vulnerable to severe COVID-19 course. Therefore, as a population, MS patients should be protected from SARS-CoV-2 infection.

Patients treated with IFN-β, teriflunomide, dimethyl fumarate, GA and natalizumab should have normal SARS-CoV-2 vaccine response [157]. It is similar with patients after post immune system reconstitution (post alemtuzumab, cladribine, mitoxantrone or hematopoietic stem cell transplantation), as long as they are not in the depletion phase of treatment [158,159,160]. Patients during the depletion phase of mentioned drugs, as well as patients treated with S1P modulators, ocrelizumab, rituximab and patients with lymphopenia during dimethyl fumarate treatment are likely to have blunted responses [158].

Three RNA vaccines against SARS-CoV-2 (Pfizer and BioNTech, Moderna, AstraZeneca) have become available since December 2020 [161]. The first two are mRNA vaccines, while AstraZeneca is a non-replicating viral vector vaccine using a live adenovirus vector. The three are not generally contraindicated during MS treatment. However, their efficacy may not be sufficient on some DMTs. More prospective data are needed. Inactivated protein-subunit (i.e., by Novavax) or inactivated virus vaccines (i.e., by Sinovac Research and Development Co., Ltd., Beijing, China) are likely to be safe and used with the same precautions as RNA vaccines in MS patients. There are only three live-attenuated SARS-CoV-2 vaccines under preclinical development- by Mehmet Ali Aydinlar University in Turkey, Codagenix and Serum Institute of India and Indian Immunologicals Ltd. and Griffith University [162]. Their potential use in MS patients is debatable.

To date it seems that the potential COVID-19 consequences outweigh the risks of vaccination. However, the literature on this specific subject remains scarce. There is only one case described of MS following SARS-CoV-2 infection [142] and several ADEM and myelitis cases [143,144,145,146,147]. The association of SARS-CoV-2 with neurological complications in general seems high, while so far, there have only been a few cases reported of neurological complications associated with vaccinations against COVID-19. These include transverse myelitis following AstraZeneca vaccine [163], 7 cases of Bell’s palsy following mRNA vaccinations [164] and a single Guillain–Barre Syndrome case [165]. Other complaints that were observed following COVID-19 vaccinations include transient and unspecific symptoms, such as headache, dizziness, muscle pain and paresthesias. Of note, 18 cases of cerebral venous sinus thrombosis were reported following AstraZeneca vaccine [166]. However, European Medicines Agency ruled that the benefits of the vaccine still outweigh the risks of side effects, which are indeed very rare but warrant further investigation.

## 3. Discussion

Physicians need to balance between the risk of infection and the risk of vaccination in a given patient; therefore, medical decisions need to be highly individualized. The exposure to a specific pathogen needs to be assessed based on the patient’s place of residence, factors associated with work and travel habits. Other issues of interest are, most importantly, the DMT the patients are on, pregnancy planning, concomitant diseases, and concomitant therapies. These factors are presented in
Figure 1.

Based on published data that we have summarized in this review, it is more likely that MS patients will suffer from the consequences of the infection rather than the potential of generating a detrimental immune response following a vaccination. Moreover, patients should undergo all the necessary vaccinations according to a vaccination calendar that is approved in their countries of residence. Moreover, it is without a doubt that MS patients need to be vaccinated against specific pathogens before specific DMTs are ordered, as advised by the most recent summaries of product characteristics; see Table 1.

By definition, live-attenuated vaccines are potentially most immunogenic. Of special note is the MMR vaccine. This should be also discussed in the context of the so-called MRZ reaction. Numerous MS patients demonstrate intrathecal polyspecific humoral immune response to many neurotropic viruses. The commonest constituents of special relevance, the Measles (M), Rubella (R) and Varicella Zoster (Z) viruses, serve as a crucial diagnostic tool [13]. This reaction against the three common neurotropic viruses, named the MRZ-reaction, is a CSF diagnostic marker that has very high specificity and positive predictive value favoring it to be a reliable “rule-in” test for MS diagnosis [167]. A positive MRZ-reaction is defined as a positive intrathecal response to at least two of the three viral agents [13].

The presence of these immune responses in CSF of MS patients and their pathophysiological role still remain unknown. A hypothesized source of an infectious cause can be strongly rebuffed since a PCR for measles, rubella and varicella zoster viruses has been shown to be negative in MRZ-reaction positive patients with MS [168]. Recent data further advocate this statement by signifying that the virus-specific fraction of total intrathecally synthesized immunoglobulins of patients with acute infections with herpes simplex virus (HSV) or measles was significantly higher (20- to 60-fold) compared to the polyspecific immune response against these antigens in MS patients [169]. With the apparent absence of an infectious cause, the MRZ-reaction is believed to represent non-specific activation of B cells within the CNS with negative viral replication [13]. However, vaccination status may influence the prevalence of the MRZ-reaction in MS and the reliability of this test in clinical practice. In a cohort study published in 2018, in MS patients who were mostly vaccinated against measles, the M reactions appeared to have the smallest involvement as a marker of the disease, in comparison to R and Z reactions [167]. This lack of positive MRZ-reaction can potentially argue that alterations in history of previous infections or immunization schedules may in fact account for these differences and should be carefully considered when implementing the MRZ-reaction into diagnostic applications.

It is imperative to prevent influenza infections among MS patients with the purpose of reducing cases of worsening MS-related symptoms and inducing new relapses. In view of IFN-β versus non-IFN-β medications, distinct considerations should be taken when deciding on type and timing of the influenza vaccine since achieving adequate seroprotection meaningfully changes from one therapeutic agent to another. Nevertheless, most drugs do not seem to reduce the immunogenic response and vaccine efficacy to the influenza vaccine.

Various MS drugs have been demonstrated to be adversely linked with the reactivation of HBV, including ofatumumab, ocrelizumab, alemtuzumab and cladribine. Serological markers for HBV should be assessed and analyzed prior to initiating these DMTs with the aim of treating patients with potent prophylactic oral antivirals. Hepatitis C virus (HCV) and human immunodeficiency virus (HIV) screening tests and detection efforts are crucial prior to introducing these DMTs as well.

Regarding the tetanus vaccine, literature came to a similar conclusion. Multiple studies demonstrated not only absence of increased risk but some have been able to illustrate favorable results with a decreased risk of MS development or relapse with tetanus vaccination. Thus, the tetanus vaccine should be recommended to MS patients when specified.

A detailed history of HPV related infections and dysplastic changes secondary to HPV infection as well as vaccination strategies against the virus are essential when deciding on introducing fingolimod therapy. In addition, encouragement by physicians to frequent surveillance efforts consisting of gynecological exams and pap smears should be strongly advised. On a different but related note, since smoking intensifies the risk of perineal and oromucosal malignancies, concomitantly aggravating MS, smoking is strongly disfavored. The HPV vaccine remains optional when initiating alemtuzumab.

Treatments targeting T cells can weaken immune response to VZV and increase the risk of HZ; therefore, it is important to test VZV status prior to such treatments. Present guidelines recommend that MS patients that lack a history of chickenpox infection and are seronegative should consider getting the varicella vaccine before starting immunosuppressive drugs. Specifically, the CDC recommends that patients taking fingolimod, alemtuzumab and cladribine should wait at least one month (4–6 weeks) after the last dose of the vaccine before initiating these drugs. The Shingrix vaccine was well tolerated among MS patients, and it was found to induce statistically significant VZV-specific immune responses.

Short and long-term benefits were clinically shown in relation to the BCG vaccine. These include decreased disease activity within the first 6 months of administration and relapse rates in CIS patients and an overall reduced risk of developing MS over a period of 5 years. Many DMTs may alter the immune system and can potentially simplify TB reactivation; however, this risk is not associated with MS patients taking IFN-β or GA, making them the safest treatments for MS patients with latent TB. Many other drugs in fact have been linked to TB reactivation cases; hence, recommendations for these patients include pre-treatment screening, BCG vaccination or even strict avoidance of these drugs, namely, ocrelizumab, alemtuzumab and cladribine.

Traveling to YF endemic countries carries a dilemma: There are conflicting data regarding potential MS exacerbations and relapse rates following the YF vaccine. Individualized, calculated and shared decisions by patients and the appropriate health care providers can be accepted when intending to travel to these areas. In a cautioned manner, each patient must consider the potential risk of exacerbation associated with the YF vaccine and weigh it against the possibility of exposure to the YFV and potentially getting infected.

Studies have widely denounced an association between MS and MMR and typhoid fever vaccination.

Nowadays, it is especially important to mention COVID-19 prevention, although long-term reliable data are definitely lacking so far. While MS patients, or any other autoimmune population, have not been included in clinical trials of COVID-19 vaccines, it is reasonable to assume that the rules and safety issues could be extrapolated from other inactivated vaccines experiences, namely influenza.

While this should be true for inactivated COVID-19 vaccines (including mRNA, non-replicating viral vector RNA, protein-based and inactivated viral), it cannot be directly applied to live attenuated vaccines. Special consideration needs to be taken with MS patients who are immunosuppressed, and several guidelines have already been published on this matter [157]. The timing of the vaccine needs to be carefully planned if MS patients are on immune therapies that may blunt the response to the vaccine, such as alemtuzumab, cladribine or ocrelizumab. On the other hand, vaccination is likely to be effective in subjects on beta-interferons, GA, dimethyl fumarate, teriflunomide and natalizumab. In patients treated with S1P modulators, such as fingolimod, especially with severe lymphopenia (which is to be expected), the response to the vaccine is likely to be adequate [157].

In conclusion, over the last years, a vast body of knowledge has been gathered in the field of immunology and vaccinology in the context of MS. While initially any vaccination was considered a potential immune trigger of exacerbation in MS, over the years, most vaccines were proven safe. Having done an extensive research of the literature on the subject, we did not find support for the hypothesis that vaccinations might trigger autoimmunity. The one exception seems to be the yellow fever vaccine, which should rather be avoided in MS patients, as there is some evidence of its association with clinical and radiological exacerbations of the disease. New technologies developed and currently used for COVID-19, namely RNA vaccines, have not been tested in autoimmune populations yet. However, with the enormous threat of the pandemics, one needs to make informed decisions in MS patients based on the available knowledge in the basics of immunology and vaccinology. An important factor to be considered is the concomitant treatment, especially if it is immunosuppressive. Each case needs to be handled and carefully analyzed individually. While in MS patients vaccinations do not seem risky after all, real-world data are needed to further support our current view.

## Figures and Tables

**Figure 1 ijms-22-03859-f001:**
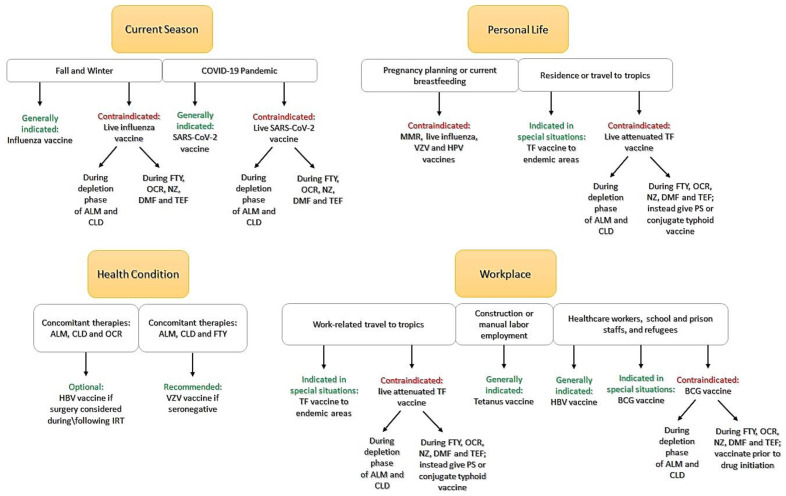
Factors that may impact individualized vaccination regimen. ALM: Alemtuzumab; CLD: Cladribine; FTY: Fingolimod; OCR: Ocrelizumab; NZ: Natalizumab; DMF: Dimethyl fumarate; TEF: Teriflunomide; COVID-19: coronavirus disease of 2019; SARS-CoV-2: severe acute respiratory syndrome coronavirus 2; MMR: measles, mumps, rubella; VZV: varicella-zoster virus; HPV: human papillomavirus; TF: Typhoid fever; PS: Polysaccharide; HBV: hepatitis B virus; IRT: immune reconstitution therapy; BCG: bacillus calmette Guerin.

**Table 1 ijms-22-03859-t001:** Vaccines and various disease modifying therapies (DMTs).

DMT	Mechanism of Action	Vaccines That Are Avoided	Recommended Vaccines Prior to Drug Initiation	Wait Period from Recommended Vaccine Administration	When to Get SARS-CoV-2 Vaccine (Pfizer BioNTech, Moderna, AstraZeneca and Janssen/J&J Vaccines)			Other
					Wait Prior to Initiating Drug	Wait after Last Regimen Given	Wait until Resuming Next Dose after Fully Vaccinated	
Alemtuzumab	Recombinant humanized monoclonal antibody to CD52	Live attenuated, including live SARS-CoV-2 vaccine during depletion phase	VZV (necessary); HPV (optional)	At least 6 weeks	4 weeks	24 weeks	More than 4 weeks	- Pap smears; - TB screen; - HBV HCV HIV screen; - PPX with anti-herpes for minimum 1 month each course of treatment
Cladribine	Deoxyadenosine purine analogue	Live attenuated, including live SARS-CoV-2 vaccine during depletion phase	VZV (necessary)	4–6 weeks	2–4 weeks	Limited data	2–4 weeks	- TB screen; - HBV HCV HIV screen
Dimethyl fumarate	Activation of Nrf2 transcriptional pathways	Live attenuated	-	-	Do not delay	Do not delay	Do not delay	-
Fingolimod	Sphingosine 1-phosphate receptor modulator	Live attenuated, including live SARS-CoV-2 vaccine during depletion phase	VZV (necessary); HPV (optional)	4 weeks after vaccination for VZV	2–4 weeks	Limited data	Limited data	- Pap smears and vaccination for HPV-related malignancies
Glatiramer acetate	Immunomodulator, analogue of MBP	-	-	-	Do not delay	Do not delay	Do not delay	-
Interferons	Immunomodulator, antiviral and immunoregulatory activities	-	-	-	Do not delay	Do not delay	Do not delay	-
Natalizumab	Recombinant humanized monoclonal antibody to integrin VLA-4	Live attenuated	-	-	Do not delay	Do not delay	Do not delay	PML screening
Ocrelizumab	Recombinant humanized monoclonal antibody against CD20	Live attenuated, including live SARS-CoV-2 vaccine during depletion phase	-	-	2–4 weeks	12 weeks	More than 4 weeks	- HBV HCV HIV screen; - TB screen
Teriflunomide	Inhibition of DHODH (de novo pyrimidine synthesis)	Live attenuated	-	-	Do not delay	Do not delay	Do not delay	-

DMT: disease modifying therapy; SARS-CoV-2: severe acute respiratory syndrome coronavirus 2; VZV: varicella zoster vaccine; HPV: human papilloma vaccine; TB: tuberculosis; HBV: hepatitis B virus; HCV: hepatitis C virus; HIV: human immunodeficiency virus; PPX: prophylaxis; Nrf2: nuclear factor erythroid 2-related factor 2; MBP: myelin basic protein; VLA-4: very late antigen-4; PML: progressive multifocal leukoencephalopathy; DHODH: dihydro-orotate dehydrogenase.

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
