# Peer review of "Current Immunological and Clinical Perspective on Vaccinations in Multiple Sclerosis Patients: Are They Safe after All?"

_ijms, 2021, doi:10.3390/ijms22083859_

Round 1

Reviewer 1 Report

I find your review very well documented and written!

Author Response

Thank you very much for such a positive review. We have made sure no spelling or style mistakes are found in the final version of the manuscript.

Reviewer 2 Report

In this review, Tsur and colleagues describe impact of MS immunopathology and MS treatments on vaccination, focusing on available information on several vaccines (influenza, HBV, tetanus, HPV, MMR, VZV, TB, yellow fever, typhoid fever, BCG) and perspective considerations on SARS-CoV-2. I enjoyed reading this article. It is informative, timely and provides meaningful insights on vaccines safety and vaccination strategy in MS patients depending on their current treatment.

Overall, the manuscript provides a pretty balanced review, well written. Doesn’t prescribe, though provides review of how many different vaccines are safe in MS patients.

Minor: Considerations for COVID19 vaccines might be included in Table 1.

Author Response

Thank you very much for a positive review! We have included considerations for COVID19 vaccines in Table 1, as suggested. We revised the style and spelling so that no mistakes would be found in the final version of the manuscript.

Reviewer 3 Report

This is a timely review. In the introduction the authors cite the hypothesis that a stimulus of the immune system, e.g., a vaccine, may trigger an autoimmune disease or its exacerbation. The review partially tracks the risks of exacerbation of MS after vaccinations, and partially deviates to the theme of infections associated with therapy and their prevention by vaccination.

The introductory section on multiple sclerosis has an elementary text book character. As other current reviews on the immunopathology of multiple sclerosis it should recall that the primary antigen is unknown. The following section on vaccinations is also very basic. The general introductory classification rather than description of therapies does not include information on efficacy, which is the background for the risks discussed in the following sections. The classification of mechanisms is relevant as different mechanisms may interfere with vaccinations. The main (and best) part of the review, on vaccination for influenza, HBV, VZV, TBC, yellow fever, typhoid fever contains rather exhaustive and useful information on each vaccination, with critical discussions. The historical development of the changing attitude towards HBV is well described. The table. including the wait periods, is instructive.

I have a few concerns.

We need a brief discussion on the diagnosis of the demyelination disorders reported (e.g. after attenuated live influenza vaccine).

The discussion on the increased risk of MS exacerbations after infections including measles is not representative of the disparate results in this part of epidemiology.

The discussion on the necessary time interval between vaccinations and the cladribine courses is not up-to-date.

Attenuated influenza vaccine response was observed in ocrelizumab treated MS patients. It is difficult to be up-to-date in these matters, but several studies appeared recently on the markedly increased risk of severe Covid-19 in patients treated with CD20-depletion compared with other immunomodulatory MS drugs. The authors need to discuss whether the concept of reconstitution therapy should be widened. It may be questionable whether “depletion phase“ is applicable to rituximab?

The title asks “…safe after all?” But they seem to be rather safe for MS. Authors could sum up the few instances where some evidence supports that vaccination may be dangerous in MS per se (for the moment disregarding therapy). Did the authors find support for the hypothesis stated in introduction that vaccinations might trigger autoimmunity? How do the authors define autoimmunity?

What do we learn from the case history of the first MS patient following Covid-19? Which features will carry an increased risk from Covid-19? And from Covid-vaccinations?? Of course, there will be a discrepancy between the solid information on the vaccinations mentioned above and the rapidly developing knowledge on coronavirus infections.

The AstraZeneca vaccine is not an RNA vector vaccine.

The figure is unsystematic in its ambition to give varying practical pieces of advice, some conditions equally important for MS and non-MS persons. Maybe avoid the formula “in situation x, do vaccination y” and use “vaccination y generally indicated / indicated in special situations / contraindicated”

Author Response

We much appreciate this thorough criticism. Below you will find a point-by-point response to the comments.

This is a timely review. In the introduction, the authors cite the hypothesis that a stimulus of the immune system, e.g., a vaccine, may trigger an autoimmune disease or its exacerbation. The review partially tracks the risks of exacerbation of MS after vaccinations, and partially deviates to the theme of infections associated with therapy and their prevention by vaccination.

The introductory section on multiple sclerosis has an elementary text book character. As other current reviews on the immunopathology of multiple sclerosis it should recall that the primary antigen is unknown. The following section on vaccinations is also very basic. The general introductory classification rather than description of therapies does not include information on efficacy, which is the background for the risks discussed in the following sections. The classification of mechanisms is relevant as different mechanisms may interfere with vaccinations.

Response: The textbook character was actually our goal. IJMS readers are a diverse population. We wanted this paper to be useful not only for MS specialists, but also neurologists that are not MS-specialists, students, family practitioners, basic scientists from outside of the neuroimmunology fields. Therefore, we made the introductory part quite basic. However, according to the suggestions of the Reviewer we have addressed the concerns and modified the introduction section, as highlighted in the manuscript, especially adding a comment on autoreactive CNS-directed B and T cells and the unknown primary antigen. Also, we modified the DMT section, adding information on the efficacy and mode of action of different therapies discussed.

The main (and best) part of the review, on vaccination for influenza, HBV, VZV, TBC, yellow fever, typhoid fever contains rather exhaustive and useful information on each vaccination, with critical discussions. The historical development of the changing attitude towards HBV is well described. The table. including the wait periods, is instructive.

I have a few concerns.

We need a brief discussion on the diagnosis of the demyelination disorders reported (e.g. after attenuated live influenza vaccine).
Response: Thank you for this suggestion. The two known cases have been described in the appropriate section, as highlighted in the text.

The discussion on the increased risk of MS exacerbations after infections including measles is not representative of the disparate results in this part of epidemiology.
Response:  We carefully read our paper but in fact, we have found no mention of increased risk of MS exacerbations/relapses following measles or other infections, except for a minor comment with regards to influenza. We do discuss measles in more detail in the discussion, but this was rather in the context of the interesting aspect of MMR reaction in CSF and the potential misinterpretation of the results in patients who have been vaccinated against measles. We hope it answers the Reviewer’s question.

The discussion on the necessary time interval between vaccinations and the cladribine courses is not up-to-date.
Response: this has been modified accordingly in the text, to be more precise. Both, the latest EMA and FDA label state vaccines should be administered 4-6 weeks before cladribine course. Different US and European recommendations vary from 2 to 6 weeks.

Attenuated influenza vaccine response was observed in ocrelizumab treated MS patients.

Response: This is mentioned in paragraph 2.1: “Patients receiving ocrelizumab, who were studied in the VELOCE study, did mount a response to the influenza vaccine, although it was attenuated compared to the control groups (receiving either IFN-β or no DMT) [63]”

It is difficult to be up-to-date in these matters, but several studies appeared recently on the markedly increased risk of severe Covid-19 in patients treated with CD20-depletion compared with other immunomodulatory MS drugs.
Response: Thank you very much for this comment. We addressed this in the COVID-19 section, as highlighted in the text.

The authors need to discuss whether the concept of reconstitution therapy should be widened. It may be questionable whether “depletion phase“ is applicable to rituximab?
Response: Thank you very much for this observation. This was actually a typo. Like most, we do not consider rituximab an immune reconstitution therapy. Instead of “namely” it should say “as well as”:Patients during the depletion phase of mentioned* drugs, namely as well as patients treated with S1P modulators, ocrelizumab, rituximab and patients with lymphopenia during dimethyl fumarate treatment are likely to have blunted responses [144].”

* these are named in the previous sentence: alemtuzumab, cladribine, mitoxantrone, or hematopoietic stem cell transplantation (all considered immune reconstitution therapies)

The title asks “…safe after all?” But they seem to be rather safe for MS. Authors could sum up the few instances where some evidence supports that vaccination may be dangerous in MS per se (for the moment disregarding therapy). Did the authors find support for the hypothesis stated in introduction that vaccinations might trigger autoimmunity? How do the authors define autoimmunity?

Response:  The question in the title addresses the general opinion among MDs and MS patients that vaccinations could aggravate MS course. As the Reviewer suggested, we have expanded the summary piece of the manuscript, pointing out the evidence does not support that and that we did not find support for this hypothesis.

The triggering of autoimmunity is a bad choice of words. We modified this in the abstract, saying simply: “Questions have been raised whether in Multiple Sclerosis (MS) patients they could also stimulate autoimmunity leading to induce disease exacerbation; and, whether vaccines could possibly act as a trigger in the onset of MS in susceptible populations.”

What do we learn from the case history of the first MS patient following Covid-19? Which features will carry an increased risk from Covid-19? And from Covid-vaccinations?? Of course, there will be a discrepancy between the solid information on the vaccinations mentioned above and the rapidly developing knowledge on coronavirus infections.

Response: Thank you for this suggestion. The appropriate discussion was introduced into the COVID-19 section of the manuscript.

The AstraZeneca vaccine is not an RNA vector vaccine.
Response: Thank you for pointing this out. We have changed the description of AstraZeneca vaccine to: a non-replicating viral vector vaccine using a live adenovirus vector ( https://www.msif.org/news/2020/02/10/the-coronavirus-and-ms-what-you-need-to-know/)

The figure is unsystematic in its ambition to give varying practical pieces of advice, some conditions equally important for MS and non-MS persons. Maybe avoid the formula “in situation x, do vaccination y” and use “vaccination y generally indicated / indicated in special situations / contraindicated”
Response: Thank you for this observation. The figure has been changed as suggested by the Reviewer.

Round 2

Reviewer 3 Report

The stated aim is clearer so the structure of the manuscript is improved. The bibliography is extensive. Considering the section on multiple sclerosis I would not recommend it as an isolated review, but it may be sufficient as introduction in the present context. 

The grammar is generally correct. However, there are a few minor language errors, and in some sections the language does not flow smoothly. Sometimes the focus switches between novel and basic information

Anti-MOG antibodies are no longer believed to be an important antigen in MS. Surprisingly, the antiviral antibodies regularly found in MS are not mentioned until their later section in MMR vaccination.

The authors state that “real” neurological complications to Covid-19 vaccines were not observed. However adverse reactions were reported. This may be premature, but if this issue needs to be mentioned, even in negative terms, the authors need to add their sources.

Returning to the figure, some people may find the diverse practical information instructive. However, for a scientific journal it could be more systematic. Perhaps could every station report on both people with MS generally and on those with immunomodulatory therapy specifically. The figure would then convey the central message of the review.

Concerning the table: The structure of the column on vaccines that are avoided seems to be unfinished, should only live or attenuated Covid-19 vaccines be avoided?

It will not be obvious for the reader that Shingrix is for the prevention of shingles, not for the prevention of VZV infection.

While the authors do point out essential facts regarding MS and vaccination, it is of course impossible to perform a quality check on this large amount of information. Results may have changed from a previous systematic review (ref. 85)? The present paper could become an updated reference on vaccinations against several concurrent diseases in MS patients.

Minor detail. Reference 135 line 776 is misplaced.

The stated aim is clearer so the structure of the manuscript is improved. The bibliography is extensive. Considering the section on multiple sclerosis I would not recommend it as an isolated review, but it may be sufficient as introduction in the present context. 

The grammar is generally correct. However, there are a few minor language errors, and in some sections the language does not flow smoothly. Sometimes the focus switches between novel and basic information

Anti-MOG antibodies are no longer believed to be an important antigen in MS. Surprisingly, the antiviral antibodies regularly found in MS are not mentioned until their later section in MMR vaccination.

The authors state that “real” neurological complications to Covid-19 vaccines were not observed. However adverse reactions were reported. This may be premature, but if this issue needs to be mentioned, even in negative terms, the authors need to add their sources.

Returning to the figure, some people may find the diverse practical information instructive. However, for a scientific journal it could be more systematic. Perhaps could every station report on both people with MS generally and on those with immunomodulatory therapy specifically. The figure would then convey the central message of the review.

Concerning the table: The column on vaccines that are avoided seems to be unfinished, should only live or attenuated Covid-19 vaccines be avoided?

It will not be obvious for the reader that Shingrix is for the prevention of shingles, not for the prevention of VZV infection. All recommendations and indications should conform with those of the manufacturer.

While the authors do point out essential facts regarding MS and vaccination, it is of course impossible to perform a quality check on this large amount of information. Results may have changed from a previous systematic review (ref. 85)? The present paper could become an updated reference on vaccinations against several concurrent diseases in MS patients.

Minor detail. Reference 135 line 776 is misplaced.

Author Response

The stated aim is clearer so the structure of the manuscript is improved. The bibliography is extensive. Considering the section on multiple sclerosis I would not recommend it as an isolated review, but it may be sufficient as introduction in the present context. 

The grammar is generally correct. However, there are a few minor language errors, and in some sections the language does not flow smoothly. Sometimes the focus switches between novel and basic information.

Response: We have made our effort to read the paper carefully again to pick up some language lapses.

Anti-MOG antibodies are no longer believed to be an important antigen in MS. Surprisingly, the antiviral antibodies regularly found in MS are not mentioned until their later section in MMR vaccination.

Response: Thank you for this important comment, the anti-MOG antibodies were indeed removed from the introduction and the section on intrathecal IgG production was further modified.

The authors state that “real” neurological complications to Covid-19 vaccines were not observed. However adverse reactions were reported. This may be premature, but if this issue needs to be mentioned, even in negative terms, the authors need to add their sources.

Response: Thank you for pointing this out. We have addressed this accordingly, adding the adequate citations.

Returning to the figure, some people may find the diverse practical information instructive. However, for a scientific journal it could be more systematic. Perhaps could every station report on both people with MS generally and on those with immunomodulatory therapy specifically. The figure would then convey the central message of the review.

Response: We have made our effort to correct the figure so that it is most accurate and instructive, hoping you will be satisfied with the modified version.

Concerning the table: The structure of the column on vaccines that are avoided seems to be unfinished, should only live or attenuated Covid-19 vaccines be avoided?

Response: The table has been corrected to be read more clearly. We hope you will be satisfied with the modified version.

It will not be obvious for the reader that Shingrix is for the prevention of shingles, not for the prevention of VZV infection.

Response: This has been corrected, as suggested.

While the authors do point out essential facts regarding MS and vaccination, it is of course impossible to perform a quality check on this large amount of information. Results may have changed from a previous systematic review (ref. 85)? The present paper could become an updated reference on vaccinations against several concurrent diseases in MS patients.

Response: Thank you. Indeed, we hope this extensive review of ours will become and updated reference on vaccinations in MS patients.

Minor detail. Reference 135 line 776 is misplaced. Response: This has been corrected.